# Identification and Patient Benefit Evaluation of Machine Learning Models for Predicting 90-Day Mortality After Endovascular Thrombectomy Based on Routinely Ready Clinical Information

**DOI:** 10.3390/bioengineering12050468

**Published:** 2025-04-28

**Authors:** Andrew Tik Ho Ng, Lawrence Wing Chi Chan

**Affiliations:** 1Department of Radiology, Pamela Youde Nethersole Eastern Hospital, Hong Kong SAR, China; nth584@ha.org.hk; 2Department of Health Technology and Informatics, The Hong Kong Polytechnic University, Hong Kong SAR, China

**Keywords:** machine learning, thrombectomy, mortality

## Abstract

Endovascular thrombectomy (EVT) is regarded as the standard of care for acute ischemic stroke (AIS) patients with large vessel occlusion (LVO). However, the mortality rates for these patients remain alarmingly high. Dependable mortality prediction based on timely clinical information is of great importance. This study retrospectively reviewed 151 patients who underwent EVT at Pamela Youde Nethersole Eastern Hospital between 1 April 2017, and 31 October 2023. The primary outcome of this study was 90-day mortality after AIS. The models were developed using two feature selection approaches (model I: sequential forward feature selection, model II: sequential forward feature selection after identifying variables through univariate logistic regression) and six algorithms. Model performance was evaluated by using external validation data of 312 cases and compared with three traditional prediction scores. This study identified support vector machine (SVM) using model II as the best algorithm among the various options. Meanwhile, the Houston Intra-Arterial recanalization 2 (HIAT2) score surpassed all algorithms with an AUC of 0.717. However, most algorithms provided a greater net benefit than the traditional prediction scores. Machine learning (ML) algorithms developed with routinely available variables could offer beneficial insights for predicting mortality in AIS patients undergoing EVT.

## 1. Introduction

Stroke is one of the leading causes of death and adult disability globally [1]. In 2021, it accounted for 3126 deaths and ranked as the fourth most common cause of death in Hong Kong [2]. Acute ischemic stroke (AIS) accounted for a large proportion of strokes. While endovascular thrombectomy (EVT) is regarded as the standard of care for AIS patients with large vessel occlusion (LVO), the mortality rate for these patients remains alarmingly high.

EVT offers an effective and cost-effective treatment option for ischemic strokes resulting from LVO. However, access to EVT remains limited in many regions worldwide [3]. This limitation can be attributed to various factors, including challenges in training neuro-interventionists, the necessity for reorganization of stroke care systems and policy reforms, and the substantial costs associated with EVT [3]. Accurately predicting mortality prior to EVT is essential for optimizing resource management. Furthermore, reliable predictions can empower patients and their families to make informed decisions and assist neurologists in navigating clinical choices and medical care after the procedure [4,5]. Although several machine learning (ML) algorithms and prediction scores [6,7,8] have been developed to predict the functional outcomes of stroke following EVT, the functional independent or poor functional outcomes transiently experienced by patients within 90 days after the stroke incident do not necessarily imply the same in the long term. The 90-day mortality is an intuitive indicator that facilitates clinicians to make deliberate decisions. Thus, dependable mortality prediction based on timely clinical information is of great importance prior to EVT.

Several prediction scores have been developed to predict the functional outcomes in AIS patients. However, existing prediction scores exhibit notable limitations in their reliability for predicting mortality. Firstly, some prediction scores (including Houston Intra-Arterial recanalization Therapy (HIAT), HIAT2, and Total Health Risks In Vascular Events (THRIVE) scores) were developed to predict poor functional outcome but were not specifically designed to predict mortality. Even though some prediction scores (including the iScore, Preadmission comorbidities, Level of consciousness, Age, and Neurologic deficit (PLAN), and Predicting Early Mortality of Ischemic Stroke (PREMISE) scores) have been developed to predict mortality in patients with AIS, these scores were not specifically designed for, nor limited to, EVT. Secondly, to facilitate the widespread use of prediction scores in clinical practice, the scores must be simple and accessible for non-specialist clinicians. The scores should also incorporate only the necessary prognostic variables while offering sufficient discrimination to accurately predict clinical outcomes [9,10]. However, the classification of stroke subtypes in the iScore and the clinical stroke syndrome in the PREMISE score are based on the Trial Org 10172 in Acute Stroke Treatment (TOAST) subtype and the Oxfordshire Stroke Classification Project Criteria, respectively. Valid determination of these classifications requires specialized expertise [9]. Thirdly, the HIAT, THRIVE, iScore, PLAN, and PREMISE scores are determined exclusively by clinical variables. Although the role of radiological variables in predicting functional outcome remains unclear [7], studies have indicated that incorporating these radiological variables can improve the performance of prediction scores that rely solely on clinical data [11,12]. Fourthly, the Predicting 90-day mortality of AIS with MT (PRACTICE) score was developed to predict 90-day mortality in patients with AIS who underwent EVT. However, it requires external validation [4]. Fifthly, continuous variables like age, National Institutes of Health Stroke Scale (NIHSS) score, baseline glucose level, and Alberta Stroke Program Early CT Score (ASPECTS) are often categorized to simplify score calculations. While these simplifications aid clinical practices, they may lead to a loss of accuracy [13]. Finally, the prediction scores were derived from a simplified logistic regression (LR) model that scales or stratifies and sums selected prognostic variables into an univariable score. However, it is crucial to recognize that this model assumes linear relationships between the variables and the log odds of the outcome. Furthermore, it is vulnerable to collinearity among the variables [14,15].

This study aimed to develop a clinical tool for predicting mortality prior to EVT. Its objectives include (i) developing ML algorithms to predict the mortality of AIS patients prior to EVT and (ii) comparing these newly developed algorithms with the HIAT2, THRIVE, and PRACTICE scores.

## 2. Materials and Methods

### 2.1. Study Population

This study retrospectively reviewed 151 patients who underwent EVT at Pamela Youde Nethersole Eastern Hospital (PYNEH) (Hong Kong) between 1 April 2017 and 31 October 2023. Patients were included if they met the following criteria: (1) age 18 years or older; (2) clinically confirmed diagnosis of AIS with anterior LVO; and (3) performed non-contrast computed tomography (NCCT) brain and computed tomography angiography (CTA) of the carotid artery and circle of Willis prior to EVT.

### 2.2. Study Variables and Missing Data

This study investigated both clinical and radiological variables. All variables that could be readily collected in clinical practice were identified as independent predictors of poor functional outcome and mortality in AIS patients undergoing EVT in previous studies. The clinical variables encompassed age, sex, baseline NIHSS score, admission systolic blood pressure, admission glucose level, history of previous stroke, atrial fibrillation, hypertension, diabetes mellitus, and previous intravenous thrombolysis (IVT) treatment. All demographic information was obtained from electronic patient records. The radiological variables included the ASPECTS and the site of occlusion. The ASPECTS score was calculated using the artificial intelligence (AI) software RAPID ASPECTS 1.0 (iSchema View, Menlo Park, CA, USA), and the occlusion site was assessed in the radiological CTA report. Missing values were imputed using Multiple Imputation by Chained Equations (MICE). Variables with over 40% missing data or patients with two missing variables were excluded from the analysis. The primary outcome of this study was 90-day mortality after AIS.

### 2.3. Statistical Analysis

To statistically evaluate each variable in relation to 90-day mortality, this study employed two methods: (i) Continuous variables were compared between the survival and mortality groups using either an independent samples *t*-test or the Mann–Whitney U test, depending on the distribution normality. (ii) Categorical variables were compared between the survival and mortality groups using either Pearson’s chi-squared test or Fisher’s exact test, based on the sample size.

This study employed online supplementary data of 312 cases (56 mortality and 256 non-mortality) from Lin et al.’s study [16] to validate the developed ML algorithms externally, as the selection criteria were similar in both datasets, except for the exclusion of unsuccessful recanalization in their study. Additionally, the performance of these algorithms was evaluated against the HIAT2, THRIVE, and PRACTICE scores. The calculation of the scores is summarized in Appendix A. The thresholds of the prediction scores were identified according to previous studies [4,17] or the maximum Youden Index if they did not have one. The performance was assessed using the area under the receiver operating characteristic curve (AUC), the area under the precision-recall curve (AUPRC), balanced accuracy, F1 score, and Matthews correlation coefficient (MCC). The calibration of the ML algorithms was assessed through the Brier score method. Comparisons of AUC among the different algorithms and prediction scores were conducted using DeLong’s test. A two-sided *p*-value of less than 0.05 was considered statistically significant. The clinical utility was evaluated using decision-curve analysis to determine the net benefits at various threshold probabilities. Additionally, the benefit of clinical decisions was defined as whether EVT would be withheld from patients who would not experience mortality within 90 days. By evaluating the 256 non-mortality cases in the external validation set, this study compared the proportion of better decisions predicted by the most effective algorithms and prediction scores to determine whether the model provided better clinical benefits to patients. The univariate analyses were performed utilizing SPSS version 29.0.2.0. The comparison among various prediction scores and algorithms was executed using MedCalc version 22.021.

### 2.4. Model Development

This study employed a range of traditional and ML algorithms, including LR, random forest (RF), extreme gradient boosting (XGB), k-nearest neighbor (KNN), support vector machine (SVM), and neural network (NN). The dataset was randomly divided into training and test sets in a 7:3 ratio. To maintain balance, an equal number of mortality and non-mortality cases were included in the test set, while the remaining cases were assigned to the training set. The Synthetic Minority Oversampling Technique (SMOTE) was applied to the training set to enhance the representation of minority classes. When datasets are imbalanced, ML algorithms tend to favor the majority class, which can lead to high overall prediction accuracy but poor results for the minority class [18]. SMOTE is a well-established method for mitigating issues related to imbalanced data by oversampling the minority class. The training data were subsequently utilized to develop and train the model. A grid search algorithm, combined with ten-fold cross-validation, was implemented to determine the optimal hyperparameters for the model. In model I, sequential forward feature selection was employed for feature selection. In contrast, the model II implemented this technique after identifying variables significantly associated with mortality at 90 days through univariate LR, setting a significance threshold of *p* < 0.05. The feature selection process in models I and II is illustrated in Figure 1. Moreover, this study incorporated Shapley Additive Explanations (SHAP) to improve interpretability. 312 AIS patients were included in the external validation set. The median age of the population was 72 years (interquartile range [IQR]: 63.75–79), and the median baseline NIHSS score was 14 (IQR: 11–18). The mortality rate at 90 days was 17.9%. A comparison of variables between patients who experienced mortality within 90 days and those who did not in the external validation set is presented in Appendix A. The algorithms were developed in Python 3.7, employing the Scikit-learn version 0.24.1 and XGBoost version 1.2.1 libraries.

## 3. Results

### 3.1. Study Population

This study comprised 162 AIS patients with anterior LVO who underwent EVT in PYNEH. Eleven patients were excluded due to missing two or more variables, leaving a total of 151 patients for analysis. Among 151 patients, 28 patients had one variable missing, including 2 for baseline NIHSS score, 6 for systolic blood pressure, and 20 for ASPECTS. The median age of this population was 78 years (IQR: 69–85), and the median baseline NIHSS score was 23 (IQR: 18.5–27). The 90-day mortality rate was found to be 25.8%.

A comparison of variables between patients who survived and those who did not at 90 days is presented in Table 1. Among the variables analyzed, significant differences were observed in age (median 84 years [IQR: 77.5–87.5] versus 75 years [IQR: 68.75–83], *p* < 0.001), baseline NIHSS score (median 25 [IQR: 20–28] versus 23 [IQR: 18–27], *p* = 0.038), admission systolic blood pressure (mean 165.51 [standard deviation (SD): 28.39] versus 153.67 [SD: 27.23], *p* = 0.022), ASPECTS (median 7 [IQR: 5–8] versus 8 [IQR: 6–9], *p* = 0.002), and internal carotid artery (ICA) or basilar artery (BA) occlusion rate (20/39 [51.3%] versus 37/112 [33%], *p* = 0.043). Additionally, there were no significant differences between the two groups in terms of gender, admission blood glucose level, history of previous stroke, presence of atrial fibrillation, hypertension, diabetes mellitus, or the use of bridging IVT.

### 3.2. Model Development

The 151 cases were randomly divided into training and test sets. The test set included 20 mortality cases and 20 non-mortality cases, randomly selected. The remaining 19 mortality cases and 92 non-mortality cases were assigned to the training set. A comparison of the demographic information and clinical characteristics of both sets is presented in Appendix A, which shows no statistically significant differences between them. To develop model I, forward sequential feature selection identified the following critical variables: age, baseline NIHSS score, atrial fibrillation, diabetes mellitus, ASPECTS, and bridging IVT. For model II, following the identification of variables with *p*-values less than 0.05 from the univariate analysis, forward sequential feature selection highlighted age, baseline NIHSS score, and ASPECTS. The lists of hyperparameters utilized and the final selected hyperparameters for all algorithms employed in models I and II are presented in Appendix A, respectively.

### 3.3. Model Performance

The AUC values for the algorithms in models I and II within the test set are summarized in Appendix A. In model I, the AUC varied from 0.675 to 0.734, with three out of six algorithms achieving an AUC greater than 0.7. KNN achieved the highest AUC among the six algorithms, while NN achieved the lowest. In model II, the AUC ranged from 0.693 to 0.744, with four out of six algorithms exceeding an AUC of 0.7. Once again, KNN achieved the highest AUC, whereas RF and XGB performed the lowest. Remarkably, five of the six algorithms using model II outperformed those using model I (excluding XGB). However, the differences were not statistically significant.

For external validation, the evaluations of the algorithms using models I and II, along with the prediction scores, are presented in Table 2. In model I, none of the algorithms achieved an AUC exceeding 0.7, with LR achieving the highest AUC of 0.691. In contrast, model II demonstrated a significant improvement, with three algorithms exceeding an AUC of 0.7: LR at 0.712, SVM at 0.705, and NN at 0.702. The HIAT2 score surpassed all others with an AUC of 0.717, making it the only prediction score to exceed the 0.7 threshold.

All AUPRC values for the algorithms and prediction scores remained below 0.5. In model I, the highest AUPRC was recorded at 0.387 for LR. Conversely, three algorithms using model II achieved AUPRCs above 0.4, with NN at 0.425, SVM at 0.421, and LR at 0.417. HIAT2 was the sole prediction score to surpass 0.4, with an AUPRC of 0.402.

Only HIAT2 achieved a balanced accuracy above 0.7, recording a score of 0.704. The highest balanced accuracy in model I was 0.606 for SVM, while in model II, it achieved 0.652 for NN. As for the F1 score, the highest in model I was 0.352 for SVM. In model II and the prediction scores, both NN and HIAT2 exceeded a score of 0.4, achieving scores of 0.418 and 0.474, respectively. Regarding the MCC, no algorithms surpassed a value of 0.3. In model I, SVM achieved the highest MCC at 0.257. In model II and the prediction scores, only RF and HIAT2 exceeded an MCC of 0.3, with scores of 0.301 and 0.340, respectively. When evaluating Brier Scores, the top three scores in model I were 0.146 for SVM, 0.150 for LR, and 0.156 for RF. The leading scores in model II were 0.143 for SVM, 0.145 for RF, and 0.149 for LR. Among the prediction scores, the Brier Scores ranked from best to worst were 0.194 for PRACTICE, 0.236 for THRIVE, and 0.292 for HIAT2.

The decision curve analyses for various algorithms on the external validation set for both models and prediction scores are illustrated in Figure 2. The benefits of clinical decisions regarding the withholding of EVT are compared between the SVM using model II and the HIAT2 scores, as shown in Table 3. In the SVM algorithm using model II, only 8.6% of patients were wrongly predicted to withhold EVT, while the HIAT2 score suggested that 23.4% of patients were likely to refrain from treatment. McNemar’s test indicated a significant result, with a chi-square value of 14.84 (95% Confidence Interval [CI]: 9.86–19.83; *p* < 0.0001).

### 3.4. Explanatory Analysis

The importance of various features in models I and II is depicted in Appendix A, respectively. In model I, age and ASPECTS were identified as the three most critical features across all six algorithms. Specifically, LR, RF, KNN, and SVM highlighted the importance of the baseline NIHSS, while XGB and NN emphasized AF. In this model, both LR and KNN ranked age as the most crucial feature, whereas RF, XGB, SVM, and NN placed greater importance on ASPECTS. In model II, LR and KNN recognized age as the most significant feature again, while the remaining algorithms opted for ASPECTS. Notably, the algorithms that designated age as the primary feature in both models (LR and KNN) achieved a higher AUC in the test set than the others. However, this trend did not persist during external validation. LR consistently achieved the highest AUC among the algorithms in both models, while KNN demonstrated the lowest AUC in external validation.

## 4. Discussion

The AUC is a widely recognized metric for assessing the performance of binary classifiers. In this study, the AUC for predicting mortality at 90 days on the test set varied between 0.675 and 0.744. These results were lower than those reported by Hoffman et al., who found AUC values of 0.78 and 0.74 for the pre-procedural RF and LR models, respectively. However, our study yielded comparable AUC values to their AUC values of 0.71 and 0.68 for the post-procedural RF and LR models, respectively [19].

Five algorithms (excluding XGB) exhibited higher AUC values in model II compared to model I, although this difference did not reach statistical significance. The superior performance in model II compared to model I was extended in the external validation, although only RF (*p =* 0.005) and NN (*p =* 0.01) demonstrated statistical significance between models I and II. Models I and II were distinguished based on their feature selection methods. This study developed model I using a wrapper method referred to as forward sequential feature selection. Conversely, model II employed a hybrid approach that integrated filter methods (specifically univariate analysis) with wrapper methods (forward sequential feature selection). The potential reason for the overall superior performance in model II could be due to the effective feature selection in the preliminary univariate LR. Although hybrid methods have the potential to enhance accuracy, decrease computational time, and simplify complexity [20], the comparison between models I and II does not convincingly demonstrate that hybrid methods provide superior predictive power compared to the wrapper method alone. Additional study is required to further explore the impact of various feature selection strategies on predicting functional outcomes following EVT in patients with AIS.

Most algorithms in models I and II (including the best model) identify ASPECTS as the most critical feature. ASPECTS is a ten-point quantitative scoring system that evaluates the extent of early ischemic changes in acute anterior circulation ischemic strokes. A lower ASPECTS score indicates increased territories of ischemic involvement, which could lead to an elevated risk of adverse outcomes. Recent guidelines from the American Heart Association/American Stroke Association (AHA/ASA) strongly recommend EVT for patients with an ASPECT of six or higher [21]. However, recent meta-analyses indicated EVT might benefit patients with ASPECTS scores below six [22,23]. This highlights the importance of developing a scoring system, which includes ASPECTS but does not consider ASPECTS alone, to predict the clinical outcomes in AIS patients.

Previous studies have also conducted external validations of the existing prediction scores regarding predicting 90-day mortality in patients with AIS undergoing EVT. These studies reported AUC values for the HIAT2 and THRIVE scores, which ranged from 0.64 to 0.77 and from 0.67 to 0.78, respectively [24,25]. In this study, the AUCs for the HIAT2 and THRIVE scores were 0.717 and 0.674, respectively, consistent with earlier research findings. In a study on mortality prediction, Hoffman et al. identified that the RF algorithm achieved an AUC of 0.78 (95% CI: 0.64–0.88), significantly outperforming both the HIAT and HIAT2 scores (*p* < 0.001 for each) [19]. This study demonstrated that the HIAT2 surpassed all algorithms using models I and II. However, this study compared the AUC by using an external validation dataset, while Hoffman et al.’s study used a test set. Moreover, the features used to generate the model were relatively simple and easier to collect in this study when compared to Hoffman et al.’s study [19].

AUPRC is generally considered a more effective metric than AUC for imbalanced datasets. This is primarily because AUPRC accounts for both precision (positive predictive value [PPV]) and recall (sensitivity), while AUC primarily focuses on sensitivity and specificity. In the context of imbalanced datasets, it is vital to accurately identify positive cases and minimize false positives, as a rise in false positives can significantly compromise precision [26]. Consequently, previous research has recommended using AUPRC for imbalanced classification challenges. However, a recent preprint article argued that AUPRC may disproportionately favor high-prevalence groups in cases of substantial class imbalance. The authors suggested utilizing AUPRC mainly in situations where the costs associated with false positives far exceed those linked to false negatives and where equity considerations are not a major concern [27].

Accuracy can often be deceptive when assessing performance on highly imbalanced datasets, as it tends to favor predictions for the majority class. Consequently, balanced accuracy is recommended for routine evaluations, as it offers a more dependable measure of performance in such contexts [28]. The F1 score is a valuable metric because it emphasizes the harmonic balance between precision and recall, making it particularly beneficial for imbalanced classes [29]. Since the AUC does not incorporate PPV or NPV (negative predictive value), the MCC is often recommended as a metric for evaluating binary classification. The MCC is a balanced measure that considers all four components of the confusion matrix in its calculation. While many regard it as a robust metric, particularly for imbalanced datasets, a study indicated that the MCC can significantly decline in performance when the dataset is imbalanced [30].

Even when algorithms exhibit strong discrimination, often quantified by the AUC, the estimated risk can still be unreliable [31]. Discrimination is commonly used to evaluate a predictive model’s performance, while calibration refers to the agreement between predicted and actual outcomes. Calibration is essential for assessing computational models in medical decision-making, diagnosis, and prognosis [32]. The Brier Score serves as a standard metric for evaluating and comparing the accuracy of binary predictions. It measures the mean squared difference between the predicted probability and the actual outcome, with a lower value being desirable [33]. It acts as a proper scoring rule, considering both discrimination and calibration. However, its inclination to favor tests with high specificity, especially in scenarios of low prevalence, limits its effectiveness in situations where high sensitivity is essential [33].

The net benefit is a decision-analytic statistic that accounts for benefits (true positives) and harms (false positives), adjusting the latter to represent their relative clinical impacts [33]. Decision Curve Analysis evaluates the net benefit of whether predictive models offer more benefits than risks. It addresses the limitations of statistical metrics and comprehensive decision-analytic methods [34]. In analyzing the decision curve, utilizing a threshold probability range of 0.1 to 0.25, the KNN and XGB algorithms using both models and the RF algorithm using model I exhibited a poor net benefit compared to other algorithms. In contrast, the LR and SVM algorithms in both models showed a slightly higher net benefit than the NN algorithm using both models and the RF algorithm using model II. Except for the KNN and XGB algorithms in both models and the RF algorithm in model I, all other algorithms provided a greater net benefit than the prediction scores.

Ultimately, this study identified the SVM using model II as the best algorithm among the various options. It consistently achieved relatively high scores across all evaluation metrics and demonstrated robust performance in the decision curve analysis. In the comparison between SVM using model II and the HIAT2 score, the findings indicated that SVM algorithms excelled in AUPRC and Brier Score. In contrast, the HIAT2 score demonstrated superior performance in AUC, balanced accuracy, F1 score, and MCC. When assessing the impact of clinical decision-making concerning the withholding of EVT, the SVM algorithm using model II resulted in 38 more patients (38/256, 14.8%) benefiting from EVT compared to the HIAT2 score. This indicated that ML algorithms could deliver enhanced clinical value by providing greater benefits relative to risks than traditional prediction scores.

This study developed a model that employed variables that were easily collected and utilized in clinical practice by non-specialists. The phrase “time is brain” emphasizes the urgent need for prompt evaluation and treatment in patients with AIS. Models requiring many variables for outcome prediction can pose challenges in emergency clinical situations. This study revealed that incorporating additional variables may not enhance predictive performance and could potentially weaken it if those variables are irrelevant. Additionally, this study provided a comprehensive assessment, integrating both statistical and clinical evaluations, of 90-day mortality predictions prior to EVT in AIS patients with LVO.

This study presents several limitations. Firstly, the small sample size might lead to instability and could significantly diminish the model’s predictive performance when applied to independent clinical populations [6]. Although this study included a minimum of 120 outcomes, adhering to the guideline of ten outcome events per predictor variable, this limitation persisted. Secondly, the model was constructed using a single-center, retrospective dataset, which might introduce selection bias and other unrecognized factors that could influence the results. While this study included external validation, most of the participants were Chinese, which might restrict the generalizability of the findings to other populations. Thirdly, despite utilizing MICE to handle missing data, the estimates derived from the available data might still be subject to bias, especially considering the relatively small dataset. Fourth, certain pre-treatment variables that have the potential to enhance model accuracy, such as the pre-stroke mRS, stroke subtype, and collateral score, were not examined in patients with AIS. Following recommendations from the AHA/ASA [21], most patients had a pre-stroke mRS of two or less in both the PYNEH and external validation datasets. Consequently, this study did not consider pre-stroke mRS as a variable. Moreover, the primary objective of this study was to develop practical ML algorithms suitable for clinical applications, which was why factors like stroke subtype and collateral score, requiring specialized expertise, were excluded. Finally, this study exclusively analyzed AIS patients with anterior LVO. It did not address those with posterior LVO, as the evaluation of the ASPECTS might differ considerably between these two groups.

## 5. Conclusions

This study developed several ML algorithms to predict 90-day mortality for AIS patients with anterior LVO prior to EVT. The performances of these ML algorithms were compared with established prediction scores. Among the various algorithms assessed, the SVM using model II, utilizing a hybrid feature selection method, emerged as the best ML model. While the AUC for this best ML model did not exceed that of the HIAT2 score, it did surpass both the THRIVE and PRACTICE scores. Clinically, several ML algorithms provided a higher net benefit when compared to the prediction scores. The best ML algorithm also achieved better decision-making outcomes, as indicated by the decision curve analysis and comparison of benefits in clinical decisions, compared to the HIAT2 score. Thus, these ML algorithms developed from routinely available variables could offer beneficial insights for predicting mortality in AIS patients undergoing EVT.

## Figures and Tables

**Figure 1 bioengineering-12-00468-f001:**
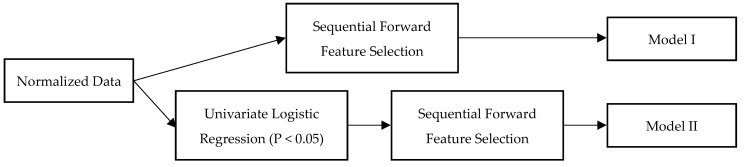
Feature selection process in models I and II.

**Figure 2 bioengineering-12-00468-f002:**
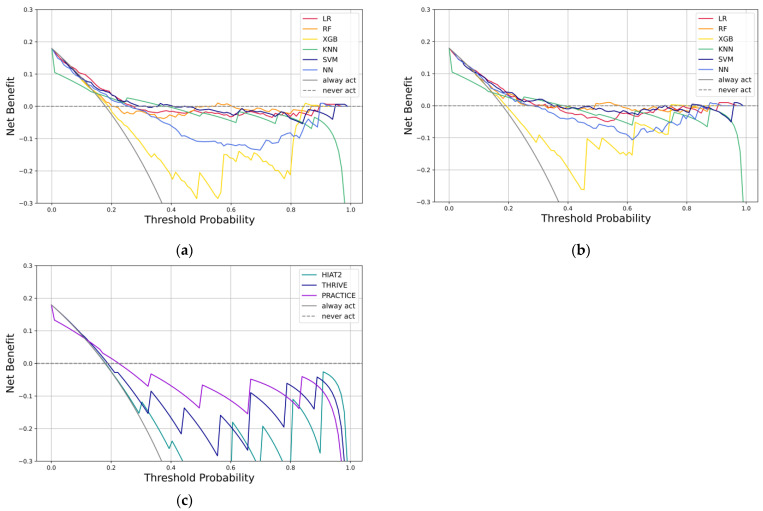
Decision Curve Analysis on the external validation set in (**a**) Model I, (**b**) Model II, and (**c**) prediction scores. HIAT: Houston Intra-Arterial Recanalization Therapy; KNN: K-Nearest Neighbor; LR: Logistic Regression; NN: Neural Network; PRACTICE: Predicting 90-day mortality of AIS with MT; RF: Random Forest; SVM: Support Vector Machine; THRIVE: Totaled Health Risks In Vascular Events; XGB: Extreme Gradient Boosting.

**Table 1 bioengineering-12-00468-t001:** Demographic and clinical characteristics between patients with and without 90-day mortality in the PYNEH data set.

Variables	Overall (n = 151)	90-Day Mortality (n = 39)	No 90-Day Mortality (n = 112)	*p*-Value
Age, years, median (IQR)	78 (69–85)	84 (77.5–87.5)	75 (68.75–83)	<0.001
Male, n (%)	57 (37.7)	14 (35.9)	43 (38.4)	0.782
Baseline NIHSS, median (IQR)	23 (18.5–27)	25 (20–28)	23 (18–27)	0.038
Admission SBP, mmHg, mean (SD)	156.73 (27.92)	165.51 (28.39)	153.67 (27.23)	0.022
Admission blood glucose level, median (IQR)	7.2 (6.2–8.9)	7.7 (6.2–9.25)	7.2 (6.175–8.75)	0.494
History of previous stroke, n (%)	34 (22.5)	9 (23.1)	25 (22.3)	0.923
Atrial Fibrillation, n (%)	78 (51.7)	22 (56.4)	56 (50)	0.490
Hypertension, n (%)	102 (67.5)	27 (69.2)	75 (67.0)	0.795
Diabetes Mellitus, n (%)	37 (24.5)	8 (20.5)	29 (25.9)	0.501
ASPECTS, median (IQR)	8 (6–9)	7 (5–8)	8 (6–9)	0.002
ICA/BA Occlusion, n (%)	57 (37.7)	20 (51.3)	37 (33.0)	0.043
Bridging IVT, n (%)	72 (47.7)	20 (51.3)	52 (46.4)	0.601

ASPECTS: Alberta Stroke Program Early CT Score; BA: Basilar Artery; ICA: Internal Carotid Artery; IQR: Interquartile Range; IVT: Intravenous Thrombolysis; NIHSS: National Institute of Health Stroke Scale; SD: Standard Deviation.

**Table 2 bioengineering-12-00468-t002:** Evaluations of the algorithms and prediction scores associated with mortality at 90 days in the external validation set. The thresholds used in the HIAT2, THRIVE, and PRACTICE scores were >6.5, >5.5, and >2.5, respectively.

Algorithm	AUC (95% CI)	AUPRC	Balanced Accuracy	F1 Score	MCC	Brier Score
LR (model I)	0.691 (0.637–0.742)	0.387	0.600	0.340	0.234	0.150
RF (model I)	0.643 (0.587–0.696)	0.327	0.596	0.329	0.253	0.156
XGB (model I)	0.606 (0.550–0.661)	0.299	0.594	0.344	0.147	0.245
KNN (model I)	0.603 (0.546–0.657)	0.328	0.558	0.244	0.161	0.163
SVM (model I)	0.668 (0.613–0.720)	0.368	0.606	0.352	0.257	0.146
NN (model I)	0.643 (0.588–0.697)	0.344	0.568	0.299	0.125	0.187
LR (model II)	0.712 (0.658–0.761)	0.417	0.609	0.358	0.228	0.149
RF (model II)	0.689 (0.634–0.740)	0.365	0.616	0.372	0.301	0.145
XGB (model II)	0.640 (0.584–0.694)	0.339	0.608	0.358	0.181	0.215
KNN (model II)	0.603 (0.547–0.658)	0.337	0.554	0.238	0.145	0.163
SVM (model II)	0.705 (0.651–0.755)	0.421	0.618	0.375	0.270	0.143
NN (model II)	0.702 (0.648–0.752)	0.425	0.652	0.418	0.270	0.167
HIAT2	0.717 (0.664–0.766)	0.402	0.704	0.474	0.340	0.292
THRIVE	0.688 (0.634–0.739)	0.351	0.589	0.331	0.158	0.236
PRACTICE	0.611 (0.554–0.665)	0.273	0.581	0.324	0.137	0.194

AUC: Area Under the Receiver Operating Characteristic Curve; AUPRC: Area Under the Precision-Recall Curve; CI: Confidence Interval; HIAT: Houston Intra-Arterial Recanalization Therapy; KNN: K-Nearest Neighbor; LR: Logistic Regression; MCC: Matthews Correlation Coefficient; NN: Neural Network; PRACTICE: Predicting 90-day mortality of AIS with MT; RF: Random Forest; SVM: Support Vector Machine; THRIVE: Totaled Health Risks In Vascular Events; XGB: Extreme Gradient Boosting.

**Table 3 bioengineering-12-00468-t003:** Comparison of the benefits of clinical decisions regarding the withholding of EVT between the SVM using model II and the HIAT2 scores by using McNemar’s test; 256 patients, who survived 90 days, were evaluated in this comparison.

	HIAT2—0	HIAT2—1		
**SVM—0**	191	43	234 (91.4%)	
**SVM—1**	5	17	22 (8.6%)	
	196 (76.6%)	60 (23.4%)	256	*p* < 0.0001

HIAT2: Houston Intra-Arterial Recanalization Therapy; SVM: Support Vector Machine.

## Data Availability

The datasets presented in this article are not readily available because the data custodian, the Hong Kong Hospital Authority, has not provided us permission. Requests to access the datasets should be directed to A.T.H.N. nth584@ha.org.hk.

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
