# Peer review of "Identification and Patient Benefit Evaluation of Machine Learning Models for Predicting 90-Day Mortality After Endovascular Thrombectomy Based on Routinely Ready Clinical Information"

_bioengineering, 2025, doi:10.3390/bioengineering12050468_

Round 1
Reviewer 1 Report
Comments and Suggestions for Authors
This study proposes novel machine learning models to predict 90-day mortality in AIS patients with anterior LVO before EVT. The experiments were conducted meticulously, and I find no significant methodological concerns. However, the concise presentation leads to insufficient explanations in several areas, making certain content difficult to understand. I propose the following improvements:
- Considering that neurosurgeons may be among the intended audience, an explanation of the Brier score method is necessary. Please clarify what the Brier score method entails and why it is essential for this study. A brief explanation of how it measures the accuracy of probabilistic predictions would benefit readers unfamiliar with this statistical approach.
- The core aspect of this research is the difference between Model I and Model II, which is currently unclear. Please provide a visual representation to clearly illustrate the differences between these two models. Additionally, further discussion on why Model II performed better would be valuable. Perhaps the preliminary univariate logistic regression step in Model II allowed for more effective feature selection compared to using sequential forward feature selection alone in Model I.
- The comparison using Decision Curve Analysis was highly beneficial. However, there is insufficient discussion of these results. The manuscript states: "In the SVM algorithm using model II, only 8.6% of patients were wrongly predicted to withhold EVT, while the HIAT2 score suggested that 23.4% of patients were likely to refrain from treatment." What specific benefits would patients receive if Model II were adopted? What is the net benefit in this Decision Curve Analysis? Even if seemingly obvious, an explanation would be necessary for readers to fully understand the clinical implications of this improvement—specifically, how reducing incorrect withholding of treatment from 23.4% to 8.6% translates to better patient outcomes in real-world clinical practice.
Author Response
Thank you very much for taking the time to review this manuscript. Please find the detailed responses below and the corresponding revisions/corrections highlighted/in track changes in the re-submitted files.
General comment: This study proposes novel machine learning models to predict 90-day mortality in AIS patients with anterior LVO before EVT. The experiments were conducted meticulously, and I find no significant methodological concerns. However, the concise presentation leads to insufficient explanations in several areas, making certain content difficult to understand. I propose the following improvements:
Comment 1: Considering that neurosurgeons may be among the intended audience, an explanation of the Brier score method is necessary. Please clarify what the Brier score method entails and why it is essential for this study. A brief explanation of how it measures the accuracy of probabilistic predictions would benefit readers unfamiliar with this statistical approach.
Response 1: Thank you for pointing this out. We agree with the comment. Therefore, we have included the explanation below.
“Discrimination is commonly used to evaluate a predictive model's performance, while calibration refers to the agreement between predicted and actual outcomes. Calibration is essential for assessing computational models in medical decision-making, diagnosis, and prognosis [32].” is added in page 9 and line 342.
Moreover, “It measures the mean squared difference between the predicted probability and the actual outcome, with the lower value being desirable [33]” is added in page 9, and line 346.
Comment 2: The core aspect of this research is the difference between Model I and Model II, which is currently unclear. Please provide a visual representation to clearly illustrate the differences between these two models. Additionally, further discussion on why Model II performed better would be valuable. Perhaps the preliminary univariate logistic regression step in Model II allowed for more effective feature selection compared to using sequential forward feature selection alone in Model I.
Response 2: Thank you for your comment. We agree with the comment. Therefore, we have added a figure to illustrate the feature selection process and add a brief discussion on why model II performed better as below.
“The feature selection process in models I and II is simply illustrated in Figure 1.” is added in page 4, and line 156.
“The potential reason for the overall superior performance in model II could be due to the effective feature selection in the preliminary univariate LR.” is added in page 8, and line 289.
Comment 3: The comparison using Decision Curve Analysis was highly beneficial. However, there is insufficient discussion of these results. The manuscript states: "In the SVM algorithm using model II, only 8.6% of patients were wrongly predicted to withhold EVT, while the HIAT2 score suggested that 23.4% of patients were likely to refrain from treatment." What specific benefits would patients receive if Model II were adopted? What is the net benefit in this Decision Curve Analysis? Even if seemingly obvious, an explanation would be necessary for readers to fully understand the clinical implications of this improvement—specifically, how reducing incorrect withholding of treatment from 23.4% to 8.6% translates to better patient outcomes in real-world clinical practice.
Response 3: Thank you for the comment. We agree with the comment. Therefore, we have added the description of net benefit as below.
“The net benefit is a decision-analytic statistic that accounts for benefits (true positives) and harms (false positives), adjusting the latter to represent their relative clinical impacts [33].” is added in page 9, and line 352.
The clinical implication of the improvement is mentioned in the manuscript as below.
“When assessing the impact of clinical decision-making concerning the withholding of EVT, the SVM algorithm using model II resulted in 38 more patients (38/256, 14.8%) benefiting from EVT compared to the HIAT2 score. This indicated that ML algorithms could deliver enhanced clinical value by providing greater benefits relative to risks than traditional prediction scores.” in page 10, and line 368.
Reviewer 2 Report
Comments and Suggestions for Authors
This paper presents a machine learning framework to predict 90-day mortality in acute ischemic stroke patients undergoing endovascular thrombectomy using routinely available clinical and radiological variables. The methodology is carefully designed, combining robust feature selection, multiple classification algorithms, and external validation with meaningful clinical relevance. The results are encouraging.
However, the rationale for using SMOTE should be more thoroughly justified, particularly with respect to potential overfitting and how its application can influence interpretability in a medical context.
Additionally, while the study reports Brier Scores and decision curve analyses, it would benefit from a short discussion on calibration quality, especially regarding the use of scikit-learn, which may provide poorly calibrated probabilities unless calibration techniques are applied.
Finally, a brief comment on the explainability of predictions for clinical deployment, i.e. complementing SHAP with feature impact at the patient level, would strengthen the manuscript.
Author Response
Thank you very much for taking the time to review this manuscript. Please find the detailed responses below and the corresponding revisions/corrections highlighted/in track changes in the re-submitted files.
General Comment: This paper presents a machine learning framework to predict 90-day mortality in acute ischemic stroke patients undergoing endovascular thrombectomy using routinely available clinical and radiological variables. The methodology is carefully designed, combining robust feature selection, multiple classification algorithms, and external validation with meaningful clinical relevance. The results are encouraging.
Comment 1: However, the rationale for using SMOTE should be more thoroughly justified, particularly with respect to potential overfitting and how its application can influence interpretability in a medical context.
Response 1: Thank you for pointing this out. We agree with the comment. Therefore, we have included the description below.
“When datasets are imbalanced, ML algorithms tend to favor the majority class, which can lead to high overall prediction accuracy but poor results for the minority class [18]. SMOTE is a well-established method for mitigating issues related to imbalanced data by oversampling the minority class.” is added in page 4, and line 147.
Comment 2: Additionally, while the study reports Brier Scores and decision curve analyses, it would benefit from a short discussion on calibration quality, especially regarding the use of scikit-learn, which may provide poorly calibrated probabilities unless calibration techniques are applied.
Response 2: Thank you for your comment. We agree with the comment. Therefore, we have added the importance of calibration below.
“Discrimination is commonly used to evaluate a predictive model's performance, while calibration refers to the agreement between predicted and actual outcomes. Calibration is essential for assessing computational models in medical decision-making, diagnosis, and prognosis [32].” is added in page 9 and line 342.
Moreover, “It measures the mean squared difference between the predicted probability and the actual outcome, with the lower value being desirable [33]” is added in page 9, and line 346.
Comment 3: Finally, a brief comment on the explainability of predictions for clinical deployment, i.e. complementing SHAP with feature impact at the patient level, would strengthen the manuscript.
Response 3: Thank you for the comment. We agree with the comment. Therefore, we have added the discussion below.
“Most algorithms in models I and II (including the best model) identify ASPECTS as the most critical feature. ASPECTS is a ten-point quantitative scoring system that evaluates the extent of early ischemic changes in acute anterior circulation ischemic strokes. A lower ASPECTS score indicates increased territories of ischemic involvement, which could lead to an elevated risk of adverse outcomes. Recent guidelines from the American Heart Association/American Stroke Association (AHA/ASA) strongly recommend EVT for patients with an ASPECT of six or higher [21]. However, recent meta-analyses indicated EVT might benefit patients with ASPECTS scores below six [22,23]. This highlights the importance of developing a scoring system, which includes ASPECTS but does not consider ASPECTS alone, to predict the clinical outcomes in AIS patients.“ is added in page 8, and line 297.